# The Overexpression of *YALI0B07117g* Results in Enhanced Erythritol Synthesis from Glycerol by the Yeast *Yarrowia lipolytica*

**DOI:** 10.3390/molecules26247549

**Published:** 2021-12-13

**Authors:** Mateusz Szczepańczyk, Dorota A. Rzechonek, Adam Dobrowolski, Aleksandra M. Mirończuk

**Affiliations:** Department of Biotechnology and Food Microbiology, Faculty of Biotechnology and Food Science, Wroclaw University of Environmental and Life Sciences, 51-630 Wroclaw, Poland; mateusz.szczepanczyk@upwr.edu.pl (M.S.); dorota.rzechonek@upwr.edu.pl (D.A.R.); adam.dobrowolski@upwr.edu.pl (A.D.)

**Keywords:** *Yarrowia lipolytica*, erythritol, glycerol, erythrose reductase, sweetener

## Abstract

The unconventional yeast *Yarrowia lipolytica* is used to produce erythritol from glycerol. In this study, the role of the erythrose reductase (ER) homolog *YALI0B07117g* in erythritol synthesis was analyzed. The deletion of the gene resulted in an increased production of mannitol (308%) and arabitol (204%) before the utilization of these polyols began. The strain overexpressing the *YALI0B07117g* gene was used to increase the erythritol yield from glycerol as a sole carbon source in batch cultures, resulting in a yield of 0.4 g/g. The specific consumption rate (qs) increased from 5.83 g/g/L for the WT strain to 8.49 g/g/L for the modified strain and the productivity of erythritol increased from 0.28 g/(L h) for the A101 strain to 0.41 g/(L h) for the modified strain. The application of the research may prove positive for shortening the cultivation time due to the increased rate of consumption of the substrate combined with the increased parameters of erythritol synthesis.

## 1. Introduction

The increase in sugar and fat consumption has led to a drastic rise in overweight and obese individuals. Obesity increases the risk of diabetes, cardiovascular diseases and certain types of cancer [1,2]. To avoid the health issues resulting from excessive sugar consumption, sugar substitutes have been implemented. Polyols, or sugar alcohols, are food additives that occur naturally in nature, especially fruit and vegetables [1,3]. The application of polyols as sweeteners became possible due to their industrial production. However, the constantly rising demand for polyols necessitates further improvement in the yield on an industrial scale.

Erythritol is a four-carbon polyol found in fruit such as pears, grapes or melons where it acts as a carbon store. Many microorganisms produce erythritol under high osmotic pressure as an osmoprotectant [4]. It is characterized by stability at high temperatures and over a wide range of pH values [5]. The popularity of erythritol is based on its low caloric value (0–0.2 kcal g^−1^) in comparison with sucrose (4 kcal g^−1^) whilst reaching approximately 60–80% of the sweetness level of sucrose [6] with a low impact on insulin release due to its chemical structure [7]. It is, therefore, safe for diabetics [8] and has antibacterial properties in relation to the creation of dental plaque [9]. Initial reports suggested that erythritol was not metabolized in the human body but a study by Hootman et al. indicated that a fraction of erythritol intake may be processed to erythronate. Moreover, the study revealed that human cells could synthetize erythritol via the pentose phosphate pathway (PPP) [10]. The industrial scale production of erythritol relies on the fermentation of *Moniliella* spp. [11] or *Candida magnoliae* [12] from glucose [3].

The oleaginous yeast *Yarrowia lipolytica* is a microorganism that is well-studied in relation to lipid synthesis [13,14,15], polyol production [16,17] and citric acid production [18,19,20]. *Y. lipolytica* possesses the ability to utilize crude glycerol, which contains many contaminants including heavy metals, methanol and salts [21], resulting in a low market price of this carbon source. Despite many years of research, the synthesis pathway for erythritol and other polyols (arabitol and mannitol) is still partially unknown and a few of the enzymes involved in the process have been cast in doubt. A recent study identified an erythrose reductase (YlER), *YALIF018590g* [22], and its role in erythritol synthesis. Additionally, 8 homologs of ER were characterized. Similarly, the d-mannitol dehydrogenase gene *YALI0B16192g* was reported by Wang et al. [23]. Erythritol metabolism remains an actively studied research area; erythritol production by *Y. lipolytica* faces problems such as a low yield and the production of byproducts. Interestingly, the yeast *Y. lipolytica* is capable of erythritol synthesis from raw glycerol [24] and its utilization [25] when the main carbon source is depleted. Therefore, this microorganism could potentially become a model in erythritol metabolism in eukaryotic cells. Further research could resolve the issues and lead to an increase of erythritol synthesis and the simultaneous decrease of byproducts from the medium.

The yeast *Y. lipolytica* is able to synthesize erythritol from glycerol as a response to high osmotic stress [4]. It was shown that the last step of the erythritol synthesis pathway is the reduction of d-erythrose to erythritol (Figure 1). This reaction is carried out by an enzyme belonging to the aldo-keto reductase family. Erythrose reductase is an enzyme responsible for catalyzing the erythrose conversion to erythritol with NADPH as an electron donor [26] following the dephosphorylation of erythrose 4-phosphate. Erythrose reductase in *Y. lipolytica* was characterized by a similarity to known enzymes from different organisms such as *Candida magnoliae* and *Moniliella megachiliensis*. The analysis of protein sequences indicated that YALI0F018590p showed a 41% and 44% similarity between the ER in the organisms mentioned above, respectively [22]. The study proposed this protein as the main ER in *Y. lipolytica*. The reported protein has been described with 8 potential homologs, including YALI0B07117p [22,27].

An in silico analysis of the potential homologs of YlER has shown a similarity between the proteins and their correlation in the phylogenic tree. To date, three potential YlER homologs have been analyzed: *YALI0F18590g* [22], *YALI0C013508g* and *YALI0D07634g* [27]. The latter two were investigated due to their dependency on NADPH as a cofactor for erythrose reductase activity. The deletion of *YALI0C013508g* and *YALI0D07634g* did not result in a significant decrease of erythritol production, leading the authors to speculate that a different protein was responsible for carrying out the reduction of erythrose to erythritol [27].

The study aim was to analyze the impact of the YlER homolog *YALIF0B07117g* as a potential way to improve the production of polyols from glycerol as a low-cost carbon source by the yeast *Y. lipolytica.* We also analyzed the influence of erythrose reductase *YALIF0B07117g* on polyol metabolisms.

## 2. Results

From among the previously described homologs of YlER, YALI0B07117p was selected for a further analysis due to the discrepancy in the N-terminal end. YALI0B07117p is the only one of the group to have a 28 amino acid sequence preceding highly similar sequences with aldo-keto reductase (AKR) domains (Figure 2). The domain and motifs search in silico yielded no results to suggest the potential role of this sequence. As the role of this sequence remains unknown, it probably does not possess any significance in erythritol synthesis. All of the homologs of ER possess highly conserved regions; one of them is the aldo-keto reductase region, which is responsible for the potential role as erythrose reductase in *Y. lipolytica*. The average size of the protein varies between 309 and 313 amino acids with two exceptions. YALI0F18590p is made up of 323 amino acids whereas YALI0B07117p is composed of 337 amino acids. The most distinguishing part of YALI0B07117p is the N-terminal end where a 28 amino acid chain is located. It does not appear in any of the other homological proteins. In all of the sequences, the conserved regions are evident. The differences between amino acids usually do not affect their properties.

In this study, the role of YALI0B07117p was analyzed in the aspect of a possible role in erythritol synthesis. To determine the effects of this protein on the metabolic pathway that results in erythritol production, the obtained strains were grown in an erythritol synthesis medium [28].

For further information about the role of *YALI0B07117g*, a knock-out mutant was obtained; the strain was named AJD ΔB07117. Shake flask experiments were performed to determine the influence of the deletion of *YALI0B07117g* on erythritol synthesis. As shown in Figure 3, the strain AJD ΔB07117 was characterized by a slower glycerol consumption compared with the control A101 strain. Initially, the knock-out strain showed no glycerol utilization on the first day of the experiment. After 72 h of the experiment, the WT strain exhausted nearly all the glycerol from the medium (Q_S_ = 1.52 g/L/h) whereas the modified strain used 90% of the supplemented carbon source (Q_S_ = 1.26 g/L/h). Erythritol synthesis in both strains remained at similar levels through the entire experiment with the wild-type strain producing 30.02 g/L (Y_ERY_ = 0.275) and the AJD ΔB07117 strain producing 28.22 g/L after 72 h (Y_ERY_ = 0.31). The final concentration of erythritol reached 29.53 g/L for AJD ΔB07117 after 96 h. The wild-type strain used erythritol as a carbon source on the last day of the experiment after the depletion of glycerol, resulting in a decrease of the erythritol level to 15.16 g/L. In addition to erythritol and glycerol, the content of the byproducts of erythritol synthesis were tested. Interestingly, the AJD ΔB07117 strain produced roughly 3 times more mannitol and 2 times more arabitol in 72 h than the control strain before the wild-type strain started to utilize these polyols (Table 1).

It could be speculated that the disruption of the erythritol synthesis pathway caused the carbon flux into mannitol and arabitol synthesis. The probable cause of this phenomenon was the maintenance of the oxide-reductase homeostasis. However, the actual mechanism of the carbon flux in polyol synthesis remains unknown.

To confirm the impact of *YALI0B07117g* on erythritol synthesis, the gene was then overexpressed under the hybrid promoter UAS1B16 [29]. Additionally, *YALI0B07117g* was overexpressed in a knock-out mutant, resulting in the complementary strain AJD-c-B07117.

First, the obtained strains AJD pAD-B07117 and AJD-c-B07117 were analyzed by a microplate reader Bioscreen C test (Oy Growth Curves Ab Ltd., Helsinki, Finland) to determine the influence of genetic modifications on their growth. The A101 strain was used as a control. The growth was measured on two carbon sources, glycerol (YNB + 5% (*w*/*v*) glycerol) (Figure 4) and glucose (YPD and YNB + 5% (*w*/*v*) glucose; data not shown). The growth curves for each experiment showed no significant difference between the modified and wild-type strains on either medium but the growth of the AJD-c-B07117 strain on glycerol was initially impaired. During the first 24 h of the experiment, the complementary strain continued to have a slower growth rate but eventually all of the strains grew to a similar OD factor by the end of 48 h. The WT strain and AJD pAD-B07117 showed similar growth on glycerol as a sole carbon source; therefore, it could be used as a control of erythritol production by the overexpression strain in shake flask and batch culture experiments where glycerol is the main carbon source.

The overexpression of *YALI0B07117g* in *Y. lipolytica* resulted in increased erythritol production in the shake flask experiments, as shown in Figure 3, compared with the control A101 strain. Additionally, the production of erythritol synthesis byproducts decreased compared with both the WT strain and the knock-out mutant (Table 1). These results suggested the potential for an increase of erythritol synthesis with glycerol as the main carbon source. Interestingly, the strain AJD pAD-B07117 showed a similar profile of glycerol consumption to the knock-out mutant. To determine the impact of *YALI0B07117g* on erythritol synthesis, the AJD pAD-B07117 strain was used for batch experiments to manage the culture conditions appropriately (mainly the pH, which has a significant influence on erythritol synthesis). The procedures were carried out as described before in an erythritol synthesis medium.

The strain AJD pAD-B07117 was characterized by a significant increase in erythritol synthesis compared with the control strain. The highest yield of erythritol was observed after 72 h (AJD pAD-B07117 Y_ERY_ = 0.4 g/g; A101 Y_ERY_ = 0.32 g/g). The yield of erythritol increased by 25% for the AJD pAD-B07117 strain. The productivity of erythritol (Q_ERY/GLY_) reached 0.41 g/(L h) for the modified strain and 0.28 g/(L h) for the WT strain, equal to a 46% increase in erythritol productivity. Both strains then used erythritol as a carbon source, resulting in a decrease of the total amount of erythritol at the end of the experiment, as shown in Figure 5.

The AJD pAD-B07117 strain utilized glycerol more efficiently than the control strain in the batch fermentation study. There was no initial impairment in the glycerol consumption compared with the shake flasks experiments. During the first day of the experiment, both strains used similar amounts of glycerol whereas after 48 h, the amount of glycerol in the modified strain dropped below 60 g/L and in the A101 strain it was more than 15 g/L higher. As a result, the Qs at this point in the experiment reached 1.95 g/(L h) in the modified strain compared with 1.52 g/(L h) for the WT strain. This feature, alongside the increased erythritol production, could shorten the batch fermentation time. During the experiment, the biomass production was analyzed. The results indicated that the biomass production for the modified strain was lower than the control; the biomass reached 17.6 g/L for the AJD pAD-B07117 strain and 22.2 g/L for A101 at the peak of erythritol synthesis. That observation correlated with the overall specific consumption rate (qs). The average calculated qs for A101 was 5.83 g/g/L whereas the modified strain was 8.49 g/g/L after 72 h of the experiment. Moreover, the overexpression of *YALI0B07117g* resulted in a decrease of mannitol and arabitol production compared with the AJD ΔB07117 strain (Table 2), indicating its role in the control of the carbon flux towards erythritol synthesis.

The analysis of polyol production by the strain AJD pAD-B07117 indicated its significance in erythritol synthesis. Not only was the lack of *YALI0B07117g* responsible for the lower erythritol synthesis but it also increased the production of the byproducts of erythritol synthesis in the form of arabitol and mannitol, altering the carbon flux of glycerol utilization. Additionally, the increase of erythritol synthesis by 25% in the AJD pAD-B07117 strain compared with the wild-type strain in correlation with a higher efficiency of substrate consumption showed the potential for using this strain in erythritol synthesis on a larger scale.

## 3. Discussion

The increased consumption of glycerol by the strain AJD pAD-B07117 could shorten the cultivation time, resulting in a more efficient erythritol synthesis. The higher glycerol uptake in the modified strain resulted in the depletion of the carbon source within 72 h. Genetic modifications of the genes involved in glycerol assimilation, mainly *YALI0F00484g*-encoding glycerol kinase (GK) and *YALI0B02948g*-encoding glycerol 3-P dehydrogenase (GDH), were reported to improve glycerol uptake in *Y. lipolytica* [26]. Here, the overexpression of the erythrose reductase homolog resulted in changes in the carbon flux, a higher erythritol titer and, eventually, a higher glycerol consumption. Obtaining a strain with an overexpression of all of the mentioned genes could have a potential in large scale batch cultivations and, as a result, reduce the costs of erythritol synthesis. Further improvements to the glycerol consumption may require the implementation of fed-batch cultures to supplement the carbon source during the cultivation.

The AJD pAD-B07117 strain reached a yield of erythritol production of 0.4 whilst using glycerol as a main carbon source. A study by Cheng et al. focusing on erythrose reductase identification analyzed the erythritol production in mutants overexpressing *YALI0D07634g* and *YALI0C13508g* [27]. The strains were characterized by a yield of erythritol of 0.58 and 0.59, respectively, but the culture was conducted with glucose as the main carbon source. Additionally, the initial concentration of glucose in the medium was set at 300 g/L. A similar yield of erythritol (0.44) was obtained in the analysis of *YALI0F18590g* where glycerol was used as a carbon source [22]. The research by Janek et al. (2017) and Cheng et al. (2018) and the analysis of *YALI0B07117g* focused on endogenous targets to improve erythritol synthesis. An additional advantage of *YALI0B07117g* is the use of glycerol as a carbon source instead of glucose, which is commercially more valuable.

The strain AJD pAD-B07117 has been analyzed as a potential means to the improve commercial production of erythritol but it is worth noting that the knock-out mutant of *YALI0B07117g* produces relatively large amounts of other polyols. The discrepancy between the mannitol and arabitol production in the overexpression and knock-out mutant in comparison with the WT strain implicated the significance of the role of *YALI0B07117g* in the synthesis of these polyols. The characterization of the genes responsible for mannitol synthesis by *Y. lipolytica* [23] from glycerol is necessary not only to overcome the problem of the purification of erythritol due to the byproducts in the medium but also to decipher the mannitol synthesis pathway. The strain AJD ΔB07117 may be used to analyze the genes involved in mannitol synthesis and may become a potential asset in future research to obtain a *Y. lipolytica* strain able to produce mannitol from glycerol with a high yield and productivity. Future research should focus on crude glycerol as a carbon source for further improvements in enhancing erythritol production. However, the lack of a significant decrease of erythritol synthesis in the AJD ΔB07117 strain indicated the potential for another gene as the main erythrose reductase in the yeast *Y. lipolytica*. The high percentage of homology between all of the ER homologs may cause a problem in establishing which one of these proteins is responsible for erythritol synthesis in general. Genetic modifications resulting in the alteration of erythrose reductase genes may cause an increase of the expression levels of other genes in this pathway as a response to osmotic stress, masking the potential influence of ER gene knock-outs.

## 4. Materials and Methods

### 4.1. Strains, Media and Culture Conditions

The strains used in this study are shown in Table 3. All of the strains belonged to the Department of Biotechnology and Food Microbiology at Wrocław University of Environmental and Life Sciences, Poland.

*Escherichia coli* strains were grown in a Luria–Bertani medium at 37 °C. A supplementation of ampicillin (100 mg/L) was necessary to screen the transformants on media plates. The *Y. lipolytica* strains were grown at 28 °C in a YPD medium containing 1% (*w*/*v*) yeast extract, 1% (*w*/*v*) peptone and 2% (*w*/*v*) glucose. The YPD medium was used for the preparation of inoculation for the bioreactor studies. During the shake flask experiments, the yeast cultures were grown in 0.25 L baffled flasks containing 0.03 L of the medium on a rotary shaker. The parameters were set at 28 °C and 200 rpm. Erythritol synthesis was carried out in an erythritol synthesis medium containing 100 g/L of glycerol as described by Mirończuk et al. (2017).

### 4.2. Cloning and Transformation Protocols

All of the restriction enzymes, the Phusion high-fidelity DNA polymerase and the T4 DNA ligase were purchased from Thermo Scientific (USA). The reactions followed standard protocols as described by the manufacturers. A Plasmid Mini Kit, Gel Out extraction kit and Genomic Mini AX Yeast Spin kit were obtained from A&A Biotechnology (Gdańsk, Poland). The isolation of plasmid DNA, DNA from gel purification and gDNA extraction followed protocols supplied by the manufacturer.

The *E. coli* transformation followed the standard chemical protocol with a selective medium containing ampicillin to plate the *Y. lipolytica* strains, which were transformed with overexpression cassettes or a deletion cassette by the lithium acetate method. The transformations resulted in the strains AJD ΔB07117g and AJD pAD-B07117g. Additionally, the strain AJD ΔB07117 was transformed with an overexpression cassette, resulting in the strain AJD-c-B07117g.

### 4.3. Gene Disruption

To disrupt *YALI0B07117g*, a disruption cassette was obtained. First, the upstream region was amplified with primers Up-B07117-F-HindIII and Up-B07117-R-SalI, resulting in a PCR product of 1039 bp. The upstream region of the gene was cloned to the plasmid pUC-Ura. Both the region and plasmid were digested with HindIII and SalI enzymes. This resulted in a pUpB07117 vector. The 1088 bp downstream region was then amplified using the primers Down-B07117-F-NotI and Down-B07117-R-SacII. The PCR product was then cloned into the pUpB07117 vector and digested with NotI and SacII enzymes, resulting in the pΔB07117 vector. The obtained vector was used as a template for the PCR reaction with primers Up-B07117-F-HindIII and Down-B07117-R-SacII, resulting in a 3533 linear deletion cassette. Following the PCR reaction, the product was extracted by the Gel Out extraction kit. The *Y. lipolytica* strain AJD was transformed, resulting in AJD ΔB07117. A PCR reaction was performed to verify the proper integration. All of the primers used in this study are listed in Table 4.

### 4.4. Construction of the Overexpression Vectors

*YALI0B07117g* was amplified from *Y. lipolytica* gDNA with primers B07117-F-SgsI and B07117-R-NheI, resulting in a 1034 bp PCR product. It was then digested with Fast Digest restriction enzymes SgsI and NheI and cloned into the pAD UAS1B_16_-TEF promoter vector. The obtained plasmid pAD-B07117 was digested with the MssI enzyme to create a linear overexpression cassette with *Y. lipolytica* rDNA sequences for targeted integrations. *Y. lipolytica* strains AJD and AJD ΔB07117 were used for lithium acetate transformations, resulting in strains AJD pAD-B07117 and AJD-*c-B07117*, respectively.

### 4.5. Bioscreen C Analysis

The strains were grown for 72 h in 0.1 L Erlenmeyer flasks with a working volume of 0.01 L of the YNB medium with ammonium sulfate supplemented with 2% (*w*/*v*) glucose on a rotary shaker at 28 °C and 200 rpm. A total of 1 mL of each culture was then transferred for an additional 48 h culturing in the YNB medium with glucose to ensure that the yeast cells were not able to accumulate nutrients. The analysis was performed in 100-well plates in 200 µL of the YPD medium, the YNB medium supplemented with 5% (*w*/*v*) glycerol, 5% (*w*/*v*) glucose or 5% (*w*/*v*) erythritol. The cells were washed with Milli-Q water and inoculated to an OD_600_ of 0.1 in each well. Quintuple experiments were performed with a Bioscreen C system (Oy Growth Curves Ab Ltd., Helsinki, Finland). The experiment parameters were set at 28 °C under continuous agitation. The growth was monitored by measuring the optical density (OD) at 420–560 nm every 30 min for 72 h.

### 4.6. Bioreactor Studies

To perform the bioreactor studies, the yeast strains were grown in the YPD medium for 72 h in 0.3 L Erlenmeyer flasks (0.1 L working volume) on a shaker at 28 °C and 200 rpm. The cultivations were performed using an erythritol synthesis medium containing 150 g/L of glycerol as described before by Mirończuk et al. [28]. An inoculum of 0.2 L was introduced to 1.8 L of the medium in a 5 L jar bioreactor (Biostat B Plus, Göttingen, Germany), resulting in a working volume of 2 L. The batch cultures were grown at 28 °C with the stir speed set to 800 rpm. The aeration was set to 0.8 L/min. The pH was established at 3.0 and monitored automatically and adjusted to a set value by the addition of NaOH (40% *w*/*v*).

### 4.7. Analytical Methods

The samples from the shake flask experiments (300 µL) were centrifuged at 4 °C for 15 min at 15,000 rpm. The supernatant was then transferred to new Eppendorf tubes. The samples from the batch culture experiments were centrifuged at 4 °C for 5 min at 6000 rpm. A total of 1 mL of the supernatant was transferred to an Eppendorf tube for an HPLC analysis and the biomass was determined via filtration on 0.45 µm pore membranes and drying at 120 °C. The concentration of the metabolites (erythritol, mannitol, arabitol and citric acid) as well as a carbon source (glycerol) was determined by the HPLC analysis using a HyperRez Carbohydrate H^+^ Column (CarbH^+^) (Thermo Scientific, Waltham, MA, USA) coupled to a UV (λ = 210 nm) (Dionex, Sunnyvale, CA, USA) and a refractive index detector (Shodex, Tokyo, Japan). The column was eluted with 25 mM trifluoroacetic acid at 65 °C and a flow rate of 600 µL/min.

### 4.8. Calculation of the Fermentation Parameters

To control the pH in the batch studies, NaOH was required. This resulted in the dilution of the medium. To avoid a calculation mistake, the added amount of NaOH had to be taken into consideration. Therefore, two parameters were calculated: the mass yield of erythritol (Y_ERY_) and the volumetric erythritol productivity (Q_ERY/GLY_), expressed in g/L/h of the total produced erythritol. Due to glycerol being used as a carbon source, Y_ERY_ was expressed in the amount of the produced erythritol from the used glycerol (g/g). The calculations followed the formula:
Y_ERY_ = P/S.   Q_ERY/GLY_ = P/V • t
where P is the amount of erythritol in the medium (g), S is the amount of consumed glycerol (g), V is the volume of the medium after the addition of NaOH and t is the time of fermentation when the samples were collected.

## 5. Conclusions

The research indicated the role of the gene *YALI0B07117g* in erythritol synthesis. Despite the increase of erythritol synthesis parameters such as yield and titer compared with the A101 strain, one of the most valuable features of the obtained strain AJD pAD-B07117 was a higher specific consumption rate, resulting in the shortening of the cultivation time. The deletion of YALI0B07117g resulted in the increased production of mannitol and arabitol, which can help in investigating the genes responsible for the metabolic pathway of polyols. Further research should focus on other ER homologs to determine the main enzyme responsible for the last step in erythritol synthesis.

## Figures and Tables

**Figure 1 molecules-26-07549-f001:**
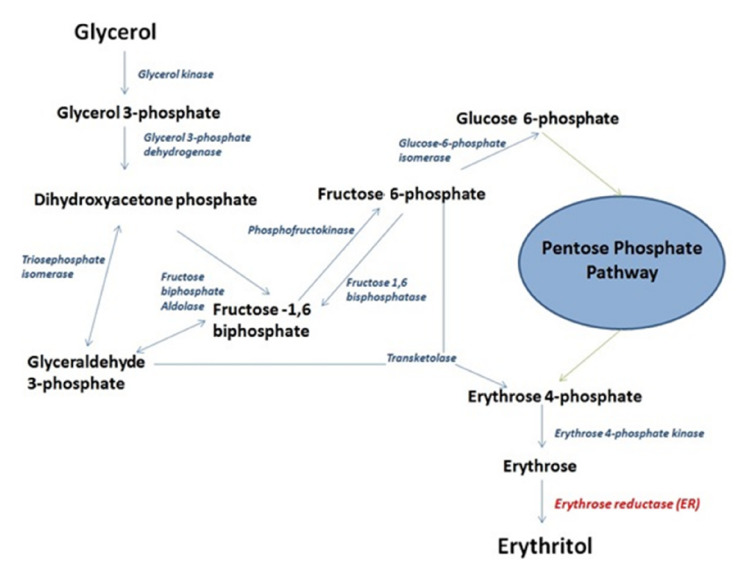
Overview of erythritol synthesis from glycerol in the yeast *Y. lipolytica*. Erythrose reductase (ER) is shown in red, other enzymes are shown in blue. The green arrows indicate the involvement in the pentose phosphate pathway. Erythrose reductase is responsible for the last reaction in the erythritol synthesis pathway, resulting in the conversion of erythrose to erythritol.

**Figure 2 molecules-26-07549-f002:**
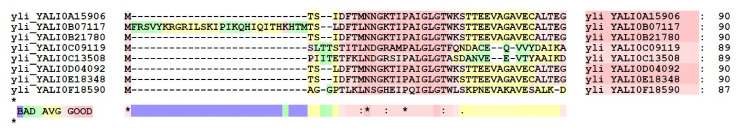
N-terminal fragment protein sequence alignment (T-COFFEE, Version 11.00). The colors indicate the level of homology: purple—low level of homology; yellow—average level of homology; red—high level of homology. The most conserved amino acids are additionally labeled with the * symbol. The lowest homology is observed within the first 30 amino acids of the N-terminal end, in which YALI0B07117g possesses a 28 amino acid sequence that is not present in other ER homologs.

**Figure 3 molecules-26-07549-f003:**
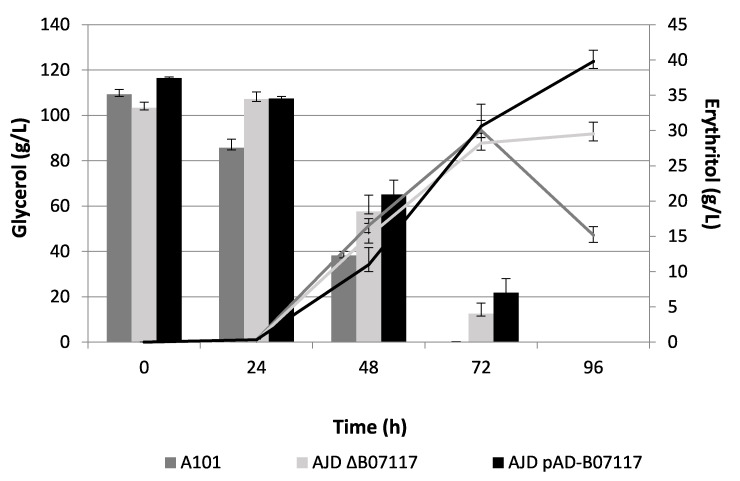
Glycerol utilization (bar chart) and erythritol production (line chart) by yeast strains A101 (gray), AJD ΔB07117 (light gray) and AJD pAD-B07117 (black) in shake flask experiments using an erythritol synthesis medium. A standard deviation was added to the chart.

**Figure 4 molecules-26-07549-f004:**
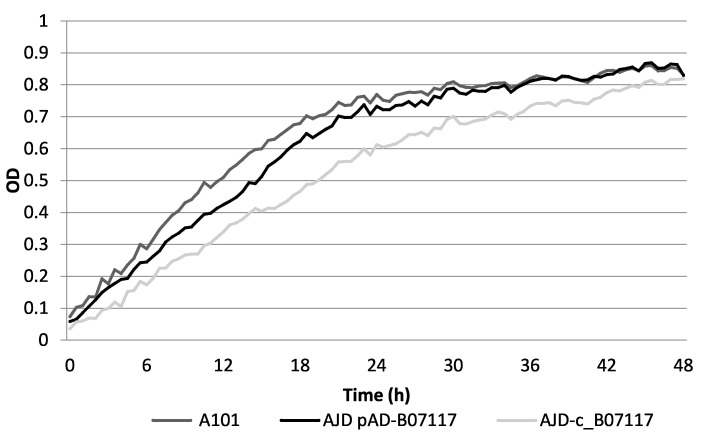
Bioscreen analysis of the strains created in this study on YNB media with 5% (*w*/*v*) glycerol. The A101 strain (gray) was used as a control. AJD pAD-B07117 (black) showed a similar growth to the control whereas AJD-c-B07117 (light grey) displayed an initially impaired growth. All of the strains reached a similar OD value by the end of the 48 h analysis.

**Figure 5 molecules-26-07549-f005:**
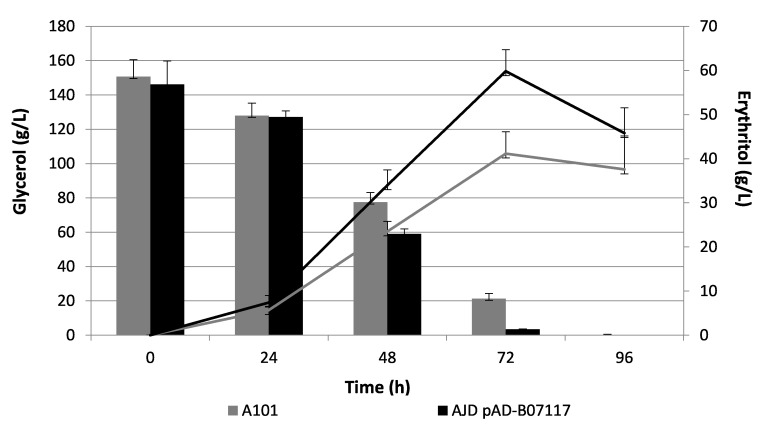
Erythritol synthesis (line chart) and glycerol utilization (bar chart) in batch reactors in an erythritol synthesis medium for the A101 strain (gray) and the AJD pAD-B07117 strain (black).

**Table 1 molecules-26-07549-t001:** Erythritol and byproduct synthesis by the *Y. lipolytica* strains in shake flask experiments.

Strain	Time (h)	Erythritol (g/L)	Mannitol (g/L)	Arabitol (g/L)	Citric Acid (g/L)	Glycerol (g/L)
A101	24	0.35 ± 0.09	0	0.01 ± 0.01	1.18 ± 0.34	87.74 ± 3.78
AJD ΔB07117		0.27 ± 0.15	0	0	0.10 ± 0.08	107.15 ± 3.2
AJD pAD-B07117		0.35 ± 0.11	0	0	0	107.4 ± 1
A101	48	16.54 ± 0.96	1.02 ± 0.16	1.61 ± 0.03	5.70 ± 0.59	38.16 ± 1.86
AJD ΔB07117		15.03 ± 1.92	4.61 ± 1.18	2.10 ± 0.46	12.04 ± 2.08	57.61 ± 7.17
AJD pAD-B07117		11.0 ± 2.4	0.13 ± 0.11	0.15 ± 0.13	8.08 ± 1.20	65.1 ± 6.39
A101	72	30.02 ± 1.39	2.43 ± 0.20	2.48 ± 0.44	5.85 ± 0.82	0.1 ± 0.18
AJD ΔB07117		28.22 ± 0.78	7.48 ± 1.49	5.07 ± 0.44	17.12 ± 3.83	12.5 ± 4.73
AJD pAD-B07117		30.60 ± 3.14	1.07 ± 0.18	0.75 ± 0.17	9.79 ± 1.60	21.78 ± 6.23
A101	96	15.16 ± 1.20	0.95 ± 0.03	1.30 ± 0.16	7.16 ± 0.67	0
AJD ΔB07117		29.52 ± 1.64	8.01 ± 1.10	6.10 ± 0.22	18.08 ± 2.87	0
AJD pAD-B07117		39.79 ± 1.60	1.00 ± 0.07	1.71 ± 0.03	9.80 ± 1.32	0

**Table 2 molecules-26-07549-t002:** Erythritol and byproduct synthesis of *Y. lipolytica* strains in bioreactor experiments.

Strain	Time (h)	Erythritol (g/L)	Mannitol (g/L)	Arabitol (g/L)	Citric Acid (g/L)	Glycerol (g/L)
A101	24	5.68 ± 0.84	0.20 ± 0.09	0.13 ± 0.05	0.84 ± 0.30	127.92 ± 7.26
AJD pAD-B07117		7.37 ± 1.62	0.05 ± 0.09	0.04 ± 0.06	0.55 ± 0.21	127.25 ± 3.46
A101	48	23.50 ± 2.27	1.59 ± 0.12	0.64 ± 0.12	2.58 ± 0.01	77.49 ± 5.62
AJD pAD-B07117		33.99 ± 3.45	2.12 ± 0.48	0.70 ± 0.06	6.20 ± 2.75	59.05 ± 2.84
A101	72	41.15 ± 4.97	3.74 ± 0.09	1.06 ± 0.11	4.57 ± 1.69	21.28 ± 3.05
AJD pAD-B07117		59.83 ± 4.86	4.72 ± 0.29	1.19 ± 0.13	12.97 ± 8.03	3.42 ± 0.18
A101	96	37.57 ± 7.59	2.09 ± 0.74	0.63 ± 0.14	9.99 ± 3.69	0.20 ± 0.35
AJD pAD-B07117		45.78 ± 5.76	4.05 ± 0.21	1.04 ± 0.07	18.60 ± 7.29	0

**Table 3 molecules-26-07549-t003:** Strains of *E. coli* and *Y. lipolytica* used in this study.

Strain	Genotype or Plasmid	Source
*E. coli*		
DH5α	F− endA1 glnV44 thi-1 recA1 relA1 gyrA96 deoR nupG Φ80dlacZΔM15 Δ(lacZYA-argF)U169, hsdR17(rK-mK+), λ–	[30]
DH5α	pUCURA	[31]
DH5α	pUCURA-B07117g	This study
DH5α	pMCSUAS_1_B_16_-TEF-lacZ	[29]
DH5α	pAD-B07117g	This study
*Y. lipolytica*		
A101	Wild-type	[32]
AJD	*MATA*, A101: ura3-302	[33]
AJD ΔB07117	*MATA*, A101: ura3-302 ΔB07117	This study
AJD-c-B07117	*MATA*, A101: ura3-302, ΔB07117 pAD-B07117	This study
AJD pAD-B07117	*MATA*, A101: ura3-302, pAD-B07117	This study

**Table 4 molecules-26-07549-t004:** Primers used in this study.

Primer	Sequence (5′ -> 3′)	Aim
YALI0B07117_up_F_HindIII	CGTAAGCTTTACACTCCCGCACAAAC	Knock-out of the *YALI0B07117g* gene
YALI0B07117_up_R_SalI	CGTGTCGACGGTCTTGCTCGGATTC	
YALI0B07117_down_F_NotI	TCAGCGGCCGCAGCTTGGTGAACCATATTT	
YALI0B07117_down_R_SacII	TATCCGCGGGCTCCAGACGAGTAAATC	
Test-YALI0B07117_F	CCCGGTTTATTGACCTCCTTACAGC	Verification of the knock-out
Test-YALI0B7117_R	CGGAACTTCTGTTTCTGCCATCTGAC	
URA_col_F	GGTACTGGTGCTTGACAGTG	
URA_col_R	CTCGAGCTAACGTCCACAAG	
YALI0B07117_F_AscI	ATAGGCGCGCCATGTTCCGGTCAGTATATAAAC	Overexpression cassette
YALI0B07117_R_NheI	GCAGCTAGCTTAGCAGAAGTCAAAGTCGGGGAAT	
TEF_F	GTCAACTCACACCCGAAATC	Verification of overexpression cassette integration
YALI0B07117_R_NheI	GCAGCTAGCTTAGCAGAAGTCAAAGTCGGGGAAT	

## Data Availability

Data are contained within the article.

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
