# Peer review of "The Overexpression of YALI0B07117g Results in Enhanced Erythritol Synthesis from Glycerol by the Yeast Yarrowia lipolytica"

_molecules, 2021, doi:10.3390/molecules26247549_

Round 1

Reviewer 1 Report

Dr. Mironczuk has good reputations in erythritol synthesis in the yeast Yarrowia lipolytica. In this manuscript, it is expected to better understand the function of YALI0B07117g. After reading the manuscript, I come up some major and minor comments.

Major comments

  1. There are eight predicted erythrose reductase homologs in Y. lipolytica. I realize it will be so difficult to study the unique function of individual homolog. I wondered if YALI0B07117g is an active erythrose reductase enzyme or not? Can this YALI0B07117g enzyme utilize other substrate(s) or only use erythose as specific substrate?

  1. Page 4, Figure 3: Can authors provide western blotting analysis result to answer the following questions.

First, in the deletion mutant, YALI0B07117g protein signal should be disappeared.

Second, in the overexpression strain, the protein expression level can be compared with wild type protein.

Third, erythrose reductase activities should be compared within the three strains.

  1. Because of eight erythrose reductase homologs, it probably has dosage compensation effect. Is it possible to monitor the protein expression levels or mRNA expression levels of the other seven homologs in the above three strains.

  1. According to page 2, line 77, three homologs were analyzed before. Is it possible to make a triple deletion mutant and quadruple deletion mutant (triple + YALI0B07117g). I am really interesting about the YALI0B07117g functions between triple and quadruple deletion mutants.

Minor comments

  1. Line 14, here is ‘qs’; but QS is used in the main text (lines115-116).

  1. Lines 37-39: Hootman et al., the reference number should be added.

  1. AJD-DB07117g was not studied in Figure 5 and Table 2, why? Line 192, the AJD-DB07117g was a typo or not?

  1. Line 216, Cheng et al (2018) is a wrong format.

  1. Table 3: Don’t separate one table on two pages.

  1. Line 313, (Mironczuk et al., 2017) is a wrong format.

  1. The whole reference format is not matched with the format of Molecules.

Reviewer 2 Report

There are minimal details of the writing style that can be improved. Some of these details have been pointed out in the attached manuscript.
In the results section, the resolution of Figure 1 should be improved, in which they show the alignment of amino acid sequences. In all figures, it is recommended that the legends be more explicit and with more details that facilitate their understanding.
While the results shown by the strain with the overexpression of the gene under study are interesting, those shown in the mutant seem a bit contradictory. It would be essential to delve into the discussion of possible causes of this behavior.

Round 2

Reviewer 1 Report

The authors answered all my questions. To me, this manuscript is acceptable.